# Metasurface-assisted massive backscatter wireless communication with commodity Wi-Fi signals

Hanting Zhao[1,4], Ya Shuang[1,4], Menglin Wei[1], Tie Jun Cui [2✉], Philipp del Hougne [3✉] & Lianlin Li [1✉]

Conventional wireless communication architecture, a backbone of our modern society, relies on actively generated carrier signals to transfer information, leading to important challenges including limited spectral resources and energy consumption. Backscatter communication systems, on the other hand, modulate an antenna's impedance to encode information into already existing waves but suffer from low data rates and a lack of information security. Here, we introduce the concept of massive backscatter communication which modulates the propagation environment of stray ambient waves with a programmable metasurface. The metasurface's large aperture and huge number of degrees of freedom enable unprecedented wave control and thereby secure and high-speed information transfer. Our prototype leveraging existing commodity 2.4 GHz Wi-Fi signals achieves data rates on the order of hundreds of Kbps. Our technique is applicable to all types of wave phenomena and provides a fundamentally new perspective on the role of metasurfaces in future wireless communication.

[1] State Key Laboratory of Advanced Optical Communication Systems and Networks, Department of Electronics, Peking University, 100871 Beijing, China. [2] State Key Laboratory of Millimeter Waves, Southeast University, 210096 Nanjing, China. [3] Univ Rennes, CNRS, Institut d'Electronique et de Télécommunications de Rennes (IETR)—UMR 6164, 35000 Rennes, France. [4] These authors contributed equally: Hanting Zhao, Ya Shuang. ✉email: tjcui@seu.edu.cn; philipp.delhougne@gmail.com; lianlin.li@pku.edu.cn

Pioneered by Guglielmo Marconi's discovery of wireless telegraphy in 1897[1], wireless communication with electromagnetic signals has become a standard approach to transfer information. Typically, a wireless communication system requires the active generation of a high-frequency carrier signal to deliver the information to be conveyed from Alice to Bob[2,3]. For this reason, we would like to refer to such schemes as active wireless communication (AWC). The capacity to transfer information with AWC is ultimately bounded by the number of available channels, for instance, how many independent spectral and spatial degrees of freedom there are available[4,5]. To meet the ever-expanding demand for more information transfer, notably given the advent of the Internet of Things (IoT), a wide range of solutions have been proposed, including elaborate coding schemes (e.g., OFDM[6]), elaborate antenna designs (e.g., massive MIMO[7]), and even the engineering of the propagation medium's disorder[8]. Nonetheless, fundamental challenges related to the need for an active carrier signal generation remain: costly and heavy hardware (oscillators, nonlinear mixers, wideband power amplifiers, etc.), power consumption, spectrum allotment issues, and information security. These issues are particularly pressing for IoT connectivity since IoT devices should be lightweight, cheap, and green.

To fundamentally address these challenges, here we take inspiration from backscatter communication systems[9–12]. The underlying idea is well known at least since the infamous Great Seal Bug[13] and has become the backbone of modern radio-frequency identification (RFID): an antenna captures a deliberately emitted signal or an opportunistic ambient signal (Wi-Fi, TV signals, 4G, etc.) and by modulating the antenna impedance, information can be encoded in the reflected and re-emitted signal. These techniques hold great promise for low-data-rate communication for RFID and IoT sensors. Nonetheless, inherent in the use of a single or few antennas is a very limited antenna aperture and number of degrees of freedom, resulting in severe restrictions on the achievable wave control. The limited wave control prevents focusing and efficient multi-channel communication, a prerequisite for transferring large amounts of information within a restricted frequency band at high speed. Moreover, from the standpoint of communication security, the limited wave control implies an inability to restrict the communication to designated target receivers, let alone to send deceiving information to an eavesdropper.

Here, we propose a massive backscatter wireless communication (MBWC) scheme in which the transmitter does not include a single or a few impedance-modulated dipole antennas but relies on modulating the propagation environment with a programmable metasurface[14,15]. We encode digital information into omnipresent stray ambient waves on the physical level using a home-made programmable metasurface in an inexpensive and dynamic manner. In contrast to existing backscatter communication schemes with a few degrees of freedom and an antenna aperture of a few cm$^2$, our proposal offers a significantly larger control over the waves. We use a 2.2 m$^2$ programmable metasurface consisting of 768 meta-atoms, such that our antenna aperture is larger by three orders of magnitude and our number of degrees of freedom is larger by several hundred orders of magnitude. The resulting unprecedented wave control enables much higher data rates and the implementation of secure communication protocols in backscatter communication while maintaining the benefits of recycling already existing waves rather than actively generating carrier signals. Our MBWC proposal opens up a promising route toward green IoT connectivity and beyond. In this article, we introduce a theoretical framework for encoding/decoding and modulation/demodulation in the proposed MBWC paradigm,

we design an inexpensive programmable metasurface operating at 2.4 GHz, and we build a proof-of-principle MBWC prototype system leveraging existing commodity 2.4 GHz Wi-Fi signals. Our experiments demonstrate secure wireless communication without any active radio components at data rates on the order of hundreds of kbps. The presented MBWC strategy provides a fundamentally new perspective on the role of programmable metasurfaces in wireless communication and can be transposed to other frequencies[16–18] and other types of wave phenomena.

## Results

**Programmable metasurface for backscatter communication.** The proposed MBWC is conceptually inspired by passive imaging schemes, which are widely utilized in the areas of radar and seismic exploration[19–23]. Passive imaging seeks to locate or detect an object of interest based on clutter signals or background noise and requires a directional reference antenna for calibrating the noncooperative illumination signal, or an antenna array for retrieving a so-called response matrix. In contrast, the proposed MBWC relies on the reallocation of ambient stray electromagnetic waves through an inexpensive programmable metasurface, as illustrated in Fig. 1a. In terms of the origin of the carrier signals our MBWC technique is thus "passive," too, but the programmable metasurface configuration is of course not static—otherwise by definition no information could be encoded. Our programmable metasurface consists of a two-dimensional (2D) array of electronically controllable digital meta-atoms[14]. For example, a 1-bit meta-atom has two digitalized states, i.e., "0" and "1," corresponding to two opposite electromagnetic responses. Hence, digital information can be assigned to the meta-atom such that it is directly encoded into the metasurface on the physical level. The power needed to program the metasurface is minimal and can be as low as a few μW per meta-atom[24]. By now, programmable metasurfaces have found various valuable applications, for instance in programmable electromagnetic imaging and sensing[25–31], wireless communication[8,32–37], dynamic holograms[38], wireless energy deposition[39,40], and analog computation with indoor Wi-Fi infrastructure[41]. The proposed MBWC paradigm utilizes the programmable metasurface for three major purposes: (1) encoding the digital information to be conveyed on the physical level; (2) directly modulating the ambient stray electromagnetic waves with high signal-to-noise ratio (SNR); and (3) facilitating the retrieval of digital information encoded into the metasurface with a matching classifier or decoder. Conceptually, our use of the metasurface to program the propagation environment of a wave with unknown characteristics (source location, angle of arrival, shape of the incident wave front) sharply differs from existing communication schemes[42] where a programmable metasurface and an active feeding-antenna source must be seen as a part of the transmit architecture rather than the propagation environment.

**MBWC system configuration.** For simplicity, we consider MBWC with binary information quantization as an illustrative example; however, the reported methods and results can be readily extended to multi-level cases. Furthermore, we assume that the information to be conveyed has been processed using standard source coding and channel coding techniques. As shown in Fig. 1b, the proposed MBWC system is composed of four major building blocks: programmable-metasurface-encoder, modulator, demodulator, and detector. Modulation/demodulation in MBWC is made with respect to an unknown stray wireless signal, as opposed to the use of prior-known local active carrier in AWC. In addition, a block diagram of AWC is summarized in Fig. 1c. To address this difficulty, we explore the unique capability

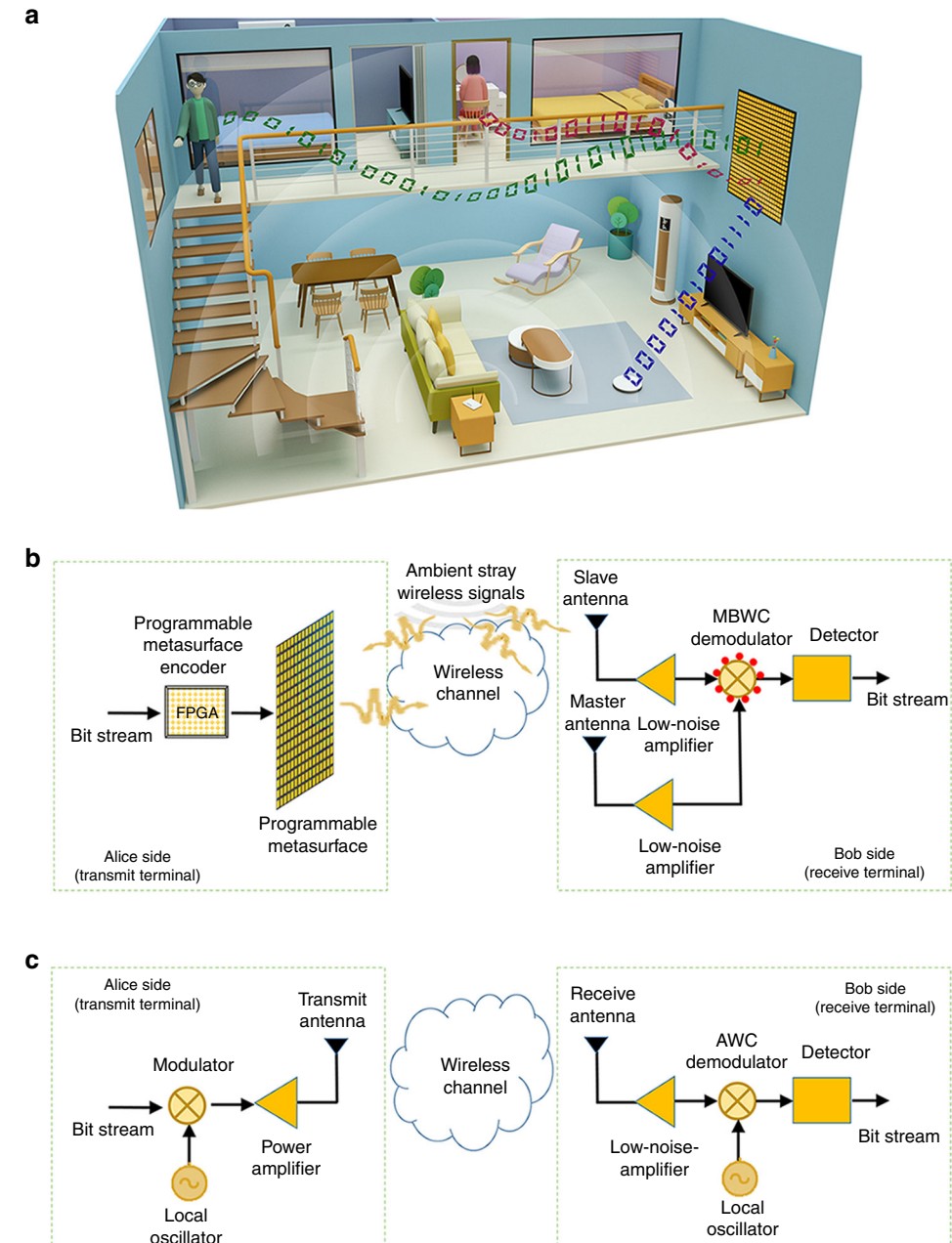

**Fig. 1 Principle of the proposed MBWC. a** An illustrative scenario for the proposed MBWC in a typical indoor environment. Here, a programmable metasurface partially decorating the wall transmits simultaneously three direction-dependent streams of digital information to three users (two people upstairs and a robot in the living room) by modulating the ambient stray electromagnetic waves. **b** Block diagram for the proposed MBWC. The programmable metasurface-based encoder maps the digital bit "0" ("1") to the metasurface configuration $\mathcal{C}_0$ ($\mathcal{C}_1$). **c** Block diagram for AWC.

of the programmable metasurface to reallocate the energy of stray wireless waves dynamically and arbitrarily toward desired spots. Using a programmable metasurface, the wireless deposition of energy at desired locations can be enhanced significantly without extra energy consumption[39]. By modulating the stray wireless signal on Alice's side, energy can be reallocated on Bob's side for demodulation provided Bob sits within the corresponding focal spot. Thereby, information can be transferred from Alice to Bob with very high SNR without generating an active carrier signal on Alice's side, and this MBWC has nearly no effect on the background AWC. Note that in order to affect the background AWC, the signal modulated by the programmable metasurface would have to be very strong at the AWC access point, which is not the case unless the AWC access point is in close vicinity of Bob's

master receiver. Similar conclusions about the negligible impact on background AWC have been reached in the literature on traditional ambient backscatter communication[10].

On Alice's side, information encoding and signal modulation are realized by the metasurface. To that end, two distinguishable digital metasurface coding patterns $\mathcal{C}_0$ and $\mathcal{C}_1$ are designed offline by solving a constrained optimization problem (see Supplementary Notes 3 and 4). For $m$-level information quantization and $k$ independent channels (or $k$ single-channel users), at least $2^{mk}$ control coding patterns are needed. "Distinguishable" here refers to the ability to distinguish the two coding patterns based on the stray wireless signals acquired and decoded or classified on Bob's side (see Fig. 2). For instance, the two control coding patterns $\mathcal{C}_0$ and $\mathcal{C}_1$ can be optimized such that the corresponding acquired

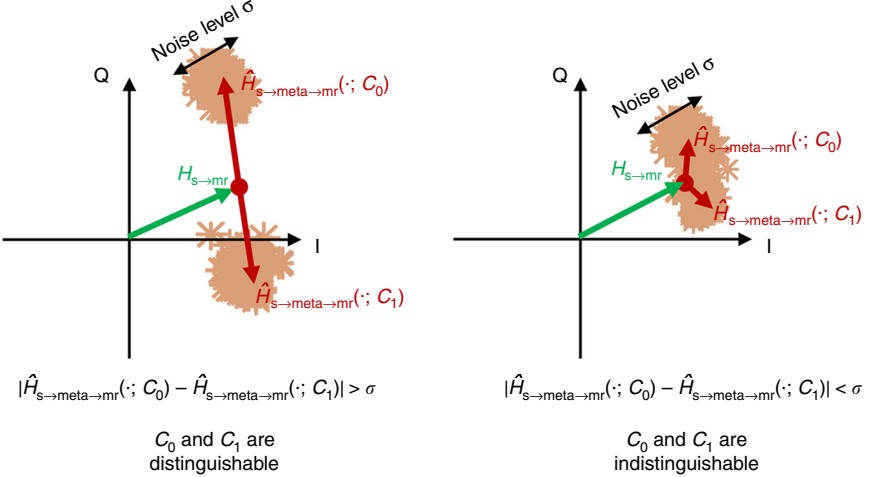

**Fig. 2 Distinguishable coding patterns in MBWC.** Constellation of the demodulated signals when the programmable metasurface is controlled by the distinguishable (left) and indistinguishable (right) coding patterns. This illustration considers binary phase shift keying modulation; other cases can be found in Supplementary Note 2.

wireless signals on Bob's side, after background removal or median filtering, have a phase difference of 180°. During operation, the sequence of digital information to be conveyed is sequentially mapped to that of the distinguishable coding patterns. For instance, the digital information "0" ("1") is mapped to $\mathcal{C}_0$ ($\mathcal{C}_1$), and thereby directly encoded into the programmable metasurface's physical layer in a time-sequential manner using a field programmable gate array. Alice's bit stream can be retrieved by analyzing the sequence of the coherence coefficients on Bob's side, in the above-mentioned example with a standard binary phase shift keying (BPSK) decoder—see details in Supplementary Note 1. The notion of distinguishability is intimately related to the SNR of the MBWC scheme: the more distinguishable the two coding patterns are, the higher is the SNR on Bob's side. The coding patterns identified by solving the constrained optimization problem ensure a high SNR for MBWC.

Retrieving the digital information from the acquired seemingly random wireless signals on Bob's side is nontrivial, since the carrier signal, i.e., the ambient stray wireless signal, is typically unknown to Alice and Bob. To address this difficulty, we deploy (at least) two coherent receivers on Bob's side to acquire the ambient stray wireless signals. The stray ambient wireless signals are only focused on one "master" receiver, which is hence "visible" to the metasurface; in contrast, a second "slave" receiver is "invisible" to the metasurface and does not capture Alice's information. The unknown modulation signal can automatically be calibrated out using multiple coherent receivers, as detailed in the next section. The focusing of the stray wireless signal on the master receiver is modulated to transfer Alice's information, yielding a high SNR demodulated signal thanks to the strong focusing ability. At the same time, the proposed MBWC scheme is expected to virtually not affect the background AWC.

It is important to note that the need for focusing on the receiver in our scheme does not inevitably mean that Bob must actively communicate his location to Alice. Instead, Alice can use a number of tricks to localize a noncooperative Bob. A first option could leverage sensor fusion by equipping the metasurface with an optical camera with depth-sensing capability (e.g., Microsoft's Kinect) such that Alice can localize Bob. In computational microwave imaging, a similar use of optical sensors to determine the region of interest has already become customary[43]. A second option could leverage the programmable metasurface for a wave-fingerprint-based position sensing scheme as described in ref. [27]

before using the metasurface for wireless communication with our MBWC proposal in a second step. A third option could rely on nonlinear feedback from Bob's receiver: many radiofrequency chains inevitably include nonlinear components, otherwise it is also possible to add such a nonlinear element on purpose. As proposed in ref. [39], based on such nonlinear feedback, Alice can blindly focus stray ambient waves on Bob's receiver—without even having estimated Bob's spatial position.

**MBWC operational principle.** For simplicity, let us assume that an unknown wireless source emits a seemingly random signal $s(t)$, and the sequence of binary digital information to be transferred from Alice to Bob is denoted by $g(\tau)$. Since the time-dependent changes in $g(\tau)$ are remarkably slower than those in $s(t)$, we refer to the time $\tau$ in $g(\tau)$ as the slow time; and the time $t$ in $s(t)$ as the fast time. In analogy with AWC, $s(t)$ and $g(\tau)$ can be correspondingly regarded as the passive carrier and the modulation signal for MBWC, respectively. We need to convert the sequence of binary information $g(\tau)$ into that of the distinguishable information-carrying control coding patterns of the programmable metasurface $\mathcal{C}(\tau)$. At any given slow time $\tau_0$, $\mathcal{C}(\tau_0) = \mathcal{C}_0$ if $g(\tau_0) = 0$ and $\mathcal{C}(\tau_0) = \mathcal{C}_1$ if $g(\tau_0) = 1$. The stray wireless signal $y_m$ acquired by the master receiver at $\mathbf{r}_m$ is the superposition of the direct arrival of the stray wireless signal and its reflection off the information-carrying metasurface, namely:

$$y_m(\tau, \mathbf{r}_m, \omega) = \widetilde{s}(\omega) H_{s \to mr}(\mathbf{r}_m, \omega) + \widetilde{s}(\omega) H_{s \to meta \to mr}(\mathbf{r}_m, \omega; \mathcal{C}(\tau)), \quad (1)$$

here $H_{s \to meta \to mr}(\mathbf{r}_m, \omega; \mathcal{C}(\tau))$ represent the response of the channel from the unknown wireless source to the master receiver via the $\mathcal{C}(\tau)$-controlled programmable metasurface, and $\widetilde{s}(\omega)$ is the frequency-domain representation of $s(t)$. In contrast, the slave receiver at $\mathbf{r}_s$ only acquires the direct arrival of stray wireless signal since it is "invisible" to the metasurface, i.e.:

$$y_s(\tau, \mathbf{r}_s, \omega) = \widetilde{s}(\omega) H_{s \to sr}(\mathbf{r}_s, \omega), \quad (2)$$

where $H_{s \to mr}$ (resp. $H_{s \to sr}$) characterizes the response of the wireless channel from the unknown source to the master (resp. slave) receiver. Then, the normalized coherence coefficient of the two acquired wireless signals reads (see Supplementary Note 1):

$$\frac{\langle y_m y_s^* \rangle}{\langle |y_s|^2 \rangle} = \widehat{H}_{s \to mr} + \widehat{H}_{s \to meta \to mr}(\mathbf{r}_m, \omega; \mathcal{C}(\tau)), \quad (3)$$

where $\widehat{H}_{s\to mr} = \frac{H_{s\to mr}}{H_{s\to sr}}$ and $\widehat{H}_{s\to meta\to mr} = \frac{H_{s\to meta\to mr}}{H_{s\to sr}}$. At this stage, three conclusions can be drawn from Eq. (3). First, $H_{s\to meta\to mr}$ as a function of $\mathcal{C}(\tau)$ carries the digital information to be conveyed by Alice, and meanwhile it is almost independent of the unknown wireless signal. Thus, we can interpret $H_{s\to meta\to mr}$ as the demodulated MBWC signal. Hence, Eq. (3) describes the MBWC demodulation and Eq. (1) the MBWC modulation procedure. Second, the terms $\widehat{H}_{s\to mr}$ and $\frac{1}{H_{s\to sr}}$ are essentially independent of the slow time $\tau$, thus $H_{s\to meta\to mr}$ can be easily retrieved by using background removal, bandpass filtering, or other simple signal processing operations. Third, the binary digital information expressed through the classes of $\mathcal{C}_i$ ($i = 0$ or 1) can be retrieved by distinguishing $H_{s\to meta\to mr}$ $(\cdot; \mathcal{C}_0)$ from $H_{s\to meta\to mr}$ $(\cdot; \mathcal{C}_1)$. As discussed in "Methods" and Supplementary Note 1, the demodulated signal $H_{s\to meta\to mr}$ reflects approximately the wavefield component radiated from the metasurface. In order to make $H_{s\to meta\to mr}$ $(\cdot; \mathcal{C}_0)$ and $H_{s\to meta\to mr}$ $(\cdot; \mathcal{C}_1)$ distinguishable in a robust way, the intensity of the signal component reflected from the metasurface needs to be comparable to that of the direct arrival from the unknown wireless source. This physical constraint can be readily realized by controlling the coding pattern of the programmable metasurface such that the energy of the stray wireless signal carrying the Alice's information is well focused and enhanced around the master receiver. In this way, $|\widehat{H}_{s\to meta\to mr}|$ can be sufficiently bigger than a decision threshold, as shown in Fig. 3d, which enables robust information retrieval. More details, for instance, for the case of distributed noncooperative wireless sources, can be found in Supplementary Notes 1 and 2.

The MBWC demodulation procedure based on the separation of time scales therefore guarantees that no AWC information (fast time $t$) is contained in the decoded MBWC signal (slow time $\tau$)—even though that information could in principle be accessed by the MBWC receiving antennas. We previously established that no MBWC information is contained in the received AWC signal since the AWC receiver is outside the focal spot in which Bob's receiver is placed. Therefore, no significant coupling between the information transferred with AWC and MBWC occurs.

Before closing this section, we briefly comment on the privacy of the presented MBWC scheme. Since Alice's information is encoded in the modulation of the focusing on the master receiver, the information cannot be retrieved at remote locations outside the focal spot (see also "Methods"). Moreover, an eavesdropper at a remote location may be deliberately deceived by Alice since the metasurface is capable of generating a number of controllable radiation beams each carrying independent streams of information. For instance, Alice can send deceitful information to an eavesdropper in an undesirable direction, which may sharply differ from the information received by Bob. MBWC will thus enable novel routes to very-high-privacy wireless communication.

**Design of information-carrying metasurface coding patterns**. The crucial component of MBWC is the programmable metasurface controlled with the distinguishable information-carrying coding patterns. We design an electronically controllable programmable metasurface, and develop an efficient optimization algorithm for finding the associated control coding patterns. For our proof-of-concept experiments, the metasurface is designed for operation around 2.4 GHz, which corresponds to the frequency range of commodity Wi-Fi signals. The designed metasurface is composed of $24 \times 32$ phase-binary reflection-type metaatoms. Each meta-atom (54 mm × 54 mm) has two binary states, "0" and "1," corresponding to two opposite reflection phases, i.e.,

0 and $\pi$, respectively, upon plane wave illumination. Switching between these two states is easily realized by changing the external DC voltage applied to the PIN diode between 3.3 and 0 V. In our design, due to fabrication constraints, the entire programmable metasurface is composed of $3 \times 4$ metasurface panels, each panel containing $8 \times 8$ meta-atoms. Each metasurface panel is equipped with eight identical 8-bit shift registers (SN74LV595APW), and the eight PIN diodes are controlled sequentially. In our design, the adopted clock (CLK) rate is 50 MHz, and the switching time of the control coding patterns of the metasurface is about 20 μs. More details on the metasurface design can be found in "Methods" and Supplementary Note 3.

As noted above, the coding patterns are optimized to have two important properties: distinguishability and "(in)visibility." For MBWC with binary information quantization, identifying the control coding patterns can be formulated as a constrained optimization problem:

$$\min_{\mathcal{C}_0, \mathcal{C}_1} \left\{ \left\| |H_{s\to meta\to mr}^{cal}(.;\mathcal{C}_0)| - |H^{des}| \right\|_2^2 + \left\| |H_{s\to meta\to mr}^{cal}(.;\mathcal{C}_1)| - |H^{des}| \right\|_2^2 \right\},$$

(4)

$$s.t., \frac{f\left(H_{s\to meta\to mr}^{cal}(.;\mathcal{C}_0)\right)}{f\left(H_{s\to meta\to mr}^{cal}(.;\mathcal{C}_1)\right)} \geq \eta.$$

As detailed in Supplementary Note 1 and "Methods," the demodulated signal $H_{s\to meta\to mr}$ approximately corresponds to the signal radiated from the programmable metasurface, such that $H_{s\to meta\to mr}^{cal}$ can be treated as the radiation beam from the metasurface. The objective function is minimized such that the information-carrying beam $|H_{s\to meta\to mr}^{cal}(\mathcal{C}_i)|$ ($i = 0$ or 1) is consistent with the desired radiation beam $|H^{des}|$, when the metasurface is controlled by the pattern $\mathcal{C}_i$. The desired beam $|H^{des}|$ is "visible" to (focused on) the master receiver, while being "invisible" to (not focused on) the slave receiver. In addition, the inequality constraint in Eq. (4) explicitly imposes the property that the adopted information-carrying coding patterns of $\mathcal{C}_0$ and $\mathcal{C}_1$ are distinguishable. This constraint comes from the famous Neyman–Pearson decision rule[44,45], where $\eta$ denotes a decision threshold, and a nonlinear function $f$ is introduced to extract the features of the input argument. More details about Eq. (4) for specific modulation schemes are provided in Supplementary Note 4.

**Experimental validation of ASK in MBWC**. For a first proof-of-concept experiment in our indoor laboratory environment, we consider MBWC with binary amplitude-shift keying (BASK) (de) modulation by leveraging stray Wi-Fi signals. Unlike AWC-BASK, MBWC-BASK means that the demodulated signal's amplitude, i.e., $|\widehat{H}_{s\to meta\to mr}|$, has two digital states, "0" and "1," corresponding to the relatively high (i.e., $|\widehat{H}_{s\to meta\to mr}| > \eta$) and low energy levels ($|\widehat{H}_{s\to meta\to mr}| \leq \eta$), respectively, where $\eta$ denotes a decision threshold. To demonstrate the capability of the proposed MBWC scheme to transfer direction-dependent digital information, we consider a three-channel MBWC with three access users (Bob-R, Bob-G, Bob-B), as shown in Fig. 3a.

In an offline calibration step, we identify $2^3$ coding patterns of the programmable metasurface by solving the optimization problem in Eq. (4). As detailed in Supplementary Note 3, we approximate the propagation medium as free space. Given the line-of-sight between metasurface and master receivers, this approximation yields good results, as evidenced by the field scans presented in Supplementary Note 3, which are in good agreement with the simulated spatial distributions of $|\widehat{H}_{s\to meta\to mr}|$ at a distance of $z = 3$ m away from the metasurface (shown in Fig. 3c). In future work, refining the propagation model, e.g., with a

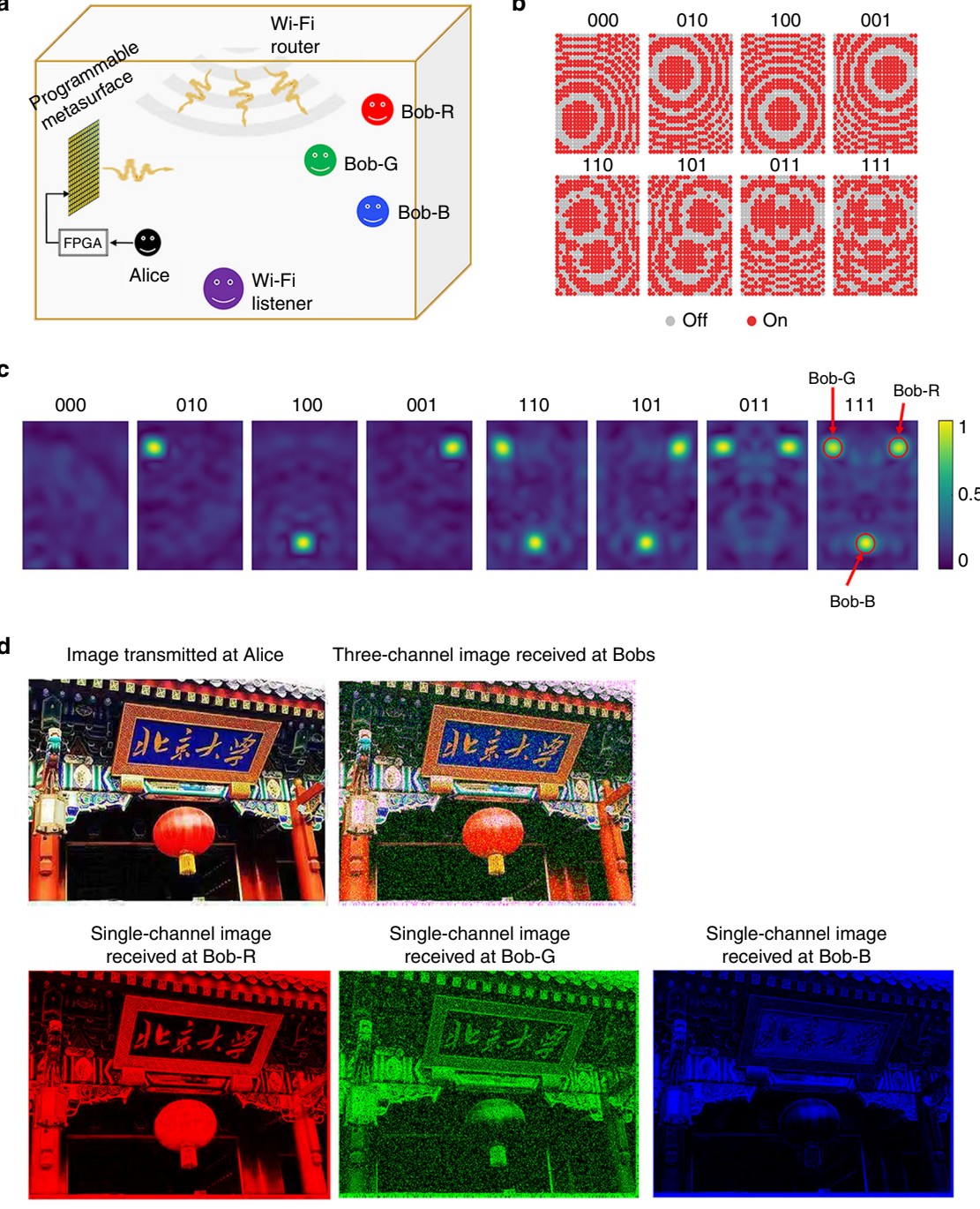

**Fig. 3 Results of the three-channel BASK MBWC. a** Schematic of the experimental setup considering a scenario with commodity Wi-Fi signals in an indoor environment. A 2.412 GHz Wi-Fi router located at an arbitrary location emits radio waves, which are captured by a standard AWC Wi-Fi user ("listener"). Alice transmits her three direction-dependent streams of information to three independent users, i.e., Bob-R (−0.86, 0, 3 m), Bob-G (0.81, 0.648, 3 m), and Bob-B (0.81, −0.648, and 3 m), by encoding the information into the programmable metasurface, and using it to modulate the stray Wi-Fi waves. The three independent users are located at a distance $z = 3$ m away from the metasurface. **b** Optimized control coding patterns for the three-channel MBWC-BASK. **c** Simulated distributions of $|H_{s \to meta \to mr}|$, corresponding to the eight coding patterns plotted in (**b**) for 2.412 GHz at the distance of $z = 3$ m from the metasurface. The intensity of stray Wi-Fi signals is well controlled for the purpose of passive amplitude modulation. **d** Individual images retrieved by Bob-R, Bob-G, and Bob-B. In addition, the full-color image transmitted by Alice, and the one synthetized using Bob's results are shown.

learned forward model[31], may further improve the performance of MBWC. During operation, we establish the programmable-metasurface-encoding scheme by mapping the three-channel binary information stream to control coding patterns of the metasurface, as reported in Fig. 3b.

Based on this MBWC information encoding scheme, we transmit a full-color image to Bob by simultaneously sending three monochrome images to Bob-R, Bob-G, and Bob-B. In our experiments, we connect four commercial electrically small antennas (one slave antenna and three master antennas) to the input ports of an oscilloscope (Agilent MSO9404A) to acquire the stray Wi-Fi signals modulated by the digital information encoded into the metasurface. The obtained results are reported in Fig. 3d.

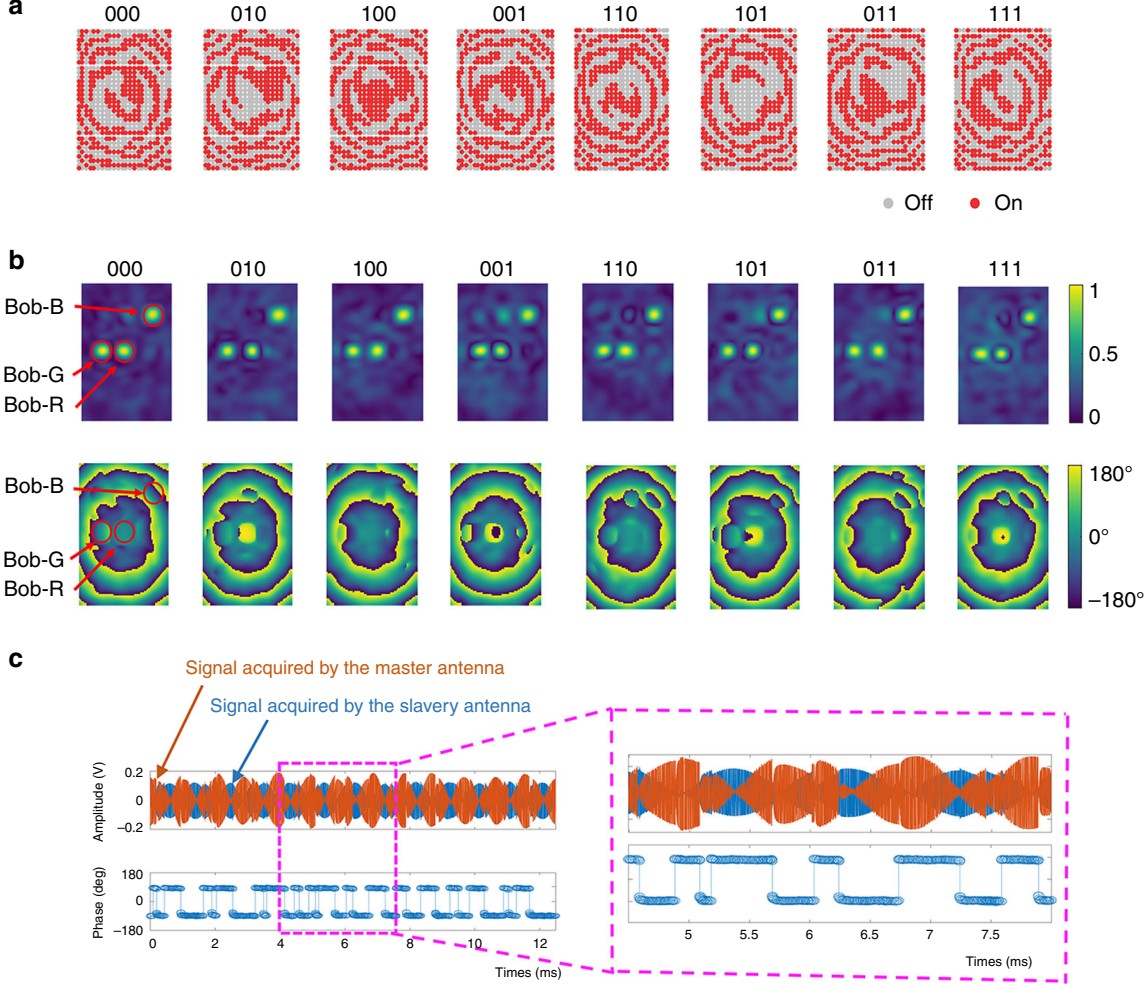

**Fig. 4 Results of the three-channel binary PSK MBWC. a** Optimized coding patterns for the three-channel MBWC-BASK. **b** Amplitude (top) and phase (bottom) distributions of $H_{s\rightarrow meta\rightarrow mr}$, corresponding to the eight coding patterns plotted in (**a**) at 2.412 GHz. Amplitude and phase of information-carrying stray Wi-Fi signals can be well controlled for the purpose of passive phase modulation. **c** (top) the stray Wi-Fi signals with about 12 ms length, which are acquired by the master (red line) and slavery (blue line) antennas, (bottom) the corresponding phase retrieved by the proposed MBWC demodulation method. Further, the signal parts within the range of 4–8 ms marked in purple are enlarged at the bottom.

We also investigate the effect of the Wi-Fi router's distance from the metasurface on MBWC. The experimental results for three additional distances (1.2, 4, and 5 m) are reported in Supplementary Note 6. The MBWC quality decreases as the router's distance increases, since the intensity of the Wi-Fi signals modulated by the metasurface, i.e., $|\widehat{H}_{s\rightarrow meta\rightarrow mr}|$, decreases correspondingly. Specifically, upon increasing the distance from 1.2 to 5 m in our experiment results in an increase of the bit error rate by 7% on average, as detailed in Supplementary Note 6. Of course, a more specialized en(de)coder can be expected to improve the communication quality. To summarize, the above-reported results evidence the ease of implementing MBWC-BASK by manipulating ambient commodity Wi-Fi signals with a carefully configured programmable metasurface.

**Experimental validation of PSK in MBWC.** As second proof-of-concept experiment, we consider MBWC with phase shift keying (PSK) (de)modulation. Unlike PSK in AWC, binary PSK here means that the calibrated coherence coefficient $\widehat{H}_{s\rightarrow meta\rightarrow mr}$ has two opposite phase states, i.e., 0 and $\pi$. Similar to above (Fig. 3a), we consider three-channel MBWC. We obtain offline eight coding patterns for controlling the metasurface by solving the constrained optimization problem in Eq. (4) using the algorithm

provided in Supplementary Note 4. Then, we build a three-channel MBWC-BPSK modulation scheme, and report the obtained results in Fig. 4a. Amplitude and phase distributions of $\widehat{H}_{s\rightarrow meta\rightarrow mr}$ at the distance of $z = 3$ m away from the metasurface are reported in Fig. 4b and Table 1. Note that the intensities of $|\widehat{H}_{s\rightarrow meta\rightarrow mr}|$ are focused around the three users, and the phases of $\widehat{H}_{s\rightarrow meta\rightarrow mr}$ are well controlled in the desired manner. Clearly, we can see that the three-channel MBWC-BPSK modulation is easily achieved by electronically switching the coding patterns of the metasurface, and that the sequence of BPSK digital information can be independently controlled for each channel. Based on these eight MBWC-BPSK coding patterns, we realize MBWC transmission of a full-color image from Alice to Bob at the distance of 3 m away from the metasurface. The corresponding experimental results are provided in Fig. 5, where the wireless signals are acquired with the four-port oscilloscope. Again, we can see that the three monochrome images can be transferred independently to three users with high communication quality.

In order to give a more realistic demonstration, we consider the transmission of a color-scale video to Bob-R, and provide experimental results in Supplementary Video 1. In this experiment, a software-defined radio (Ettus USRP X310) with two commercial electronically small antennas is utilized to acquire the

**Table 1 Experimentally measured amplitudes and phases on Bob's side for the optimized eight different coding patterns.**

| Code | Bob-R | | Bob-G | | Bob-B | |
|------|-----------|----------|-----------|----------|-----------|----------|
| | Amplitude | Phase | Amplitude | Phase | Amplitude | Phase |
| 000 | 1.50 | 5.71 | 1.55 | −16.52 | 1.52 | 3.63 |
| 100 | 1.46 | 178.12 | 1.45 | −9.37 | 1.45 | 4.10 |
| 010 | 1.56 | 6.38 | 1.56 | −173.93 | 1.56 | 0.56 |
| 110 | 1.33 | −172.98 | 1.32 | 179.56 | 1.32 | 2.58 |
| 001 | 1.38 | 11.67 | 1.38 | −20.32 | 1.38 | 176.47 |
| 101 | 1.33 | 176.46 | 1.31 | −12.38 | 1.32 | 173.29 |
| 011 | 1.43 | 5.01 | 1.43 | −168.71 | 1.44 | 175.95 |
| 111 | 1.22 | 179.04 | 1.20 | −178.68 | 1.23 | 176.95 |

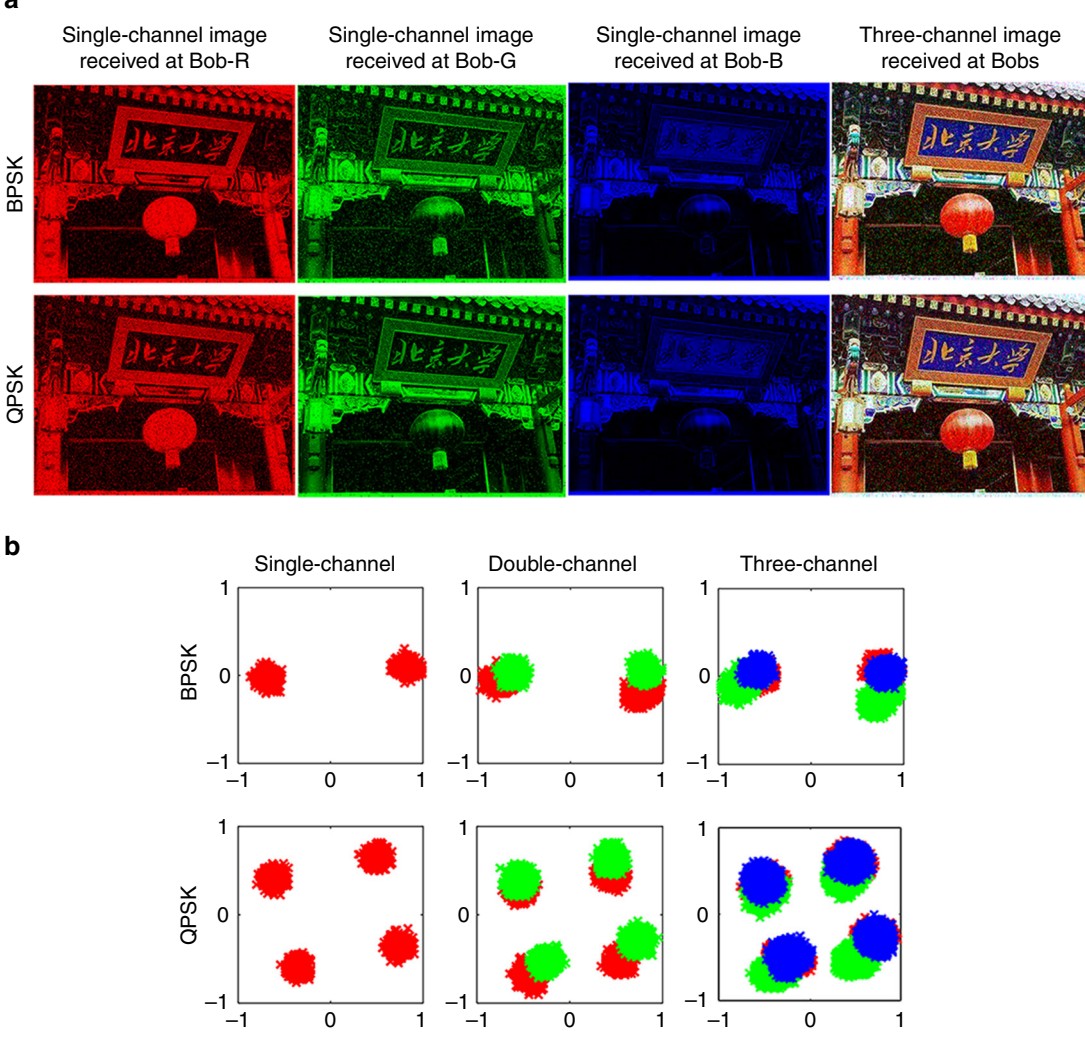

**Fig. 5 Results of the BPSK MBWC and QPSK MBWC. a** Individual monochrome and synthesized full-color images transmitted from Alice to Bob by MBWC-BPSK and MBWC-QPSK. **b** Constellation results of MBWC-BPSK and MBWC-QPSK.

stray Wi-Fi signals modulated by MBWC-BPSK. Now, we can clearly see that the information sequence encoded into the metasurface can be transferred from Alice to Bob with a data rate of 500 kbps—without actively generating a carrier signal on Alice's side nor affecting the background active Wi-Fi communication. Selected information on the fabricated demo system is provided in Fig. 4c and Table 2. More details about the demo can be found in Supplementary Note 8. It is important to note that

MBWC achieves this information transfer without using any additional spectral resources since it creates additional communication channels that make use of already existing waves from AWC. The use of MBWC has therefore the potential to help our modern society to make more efficient use of scarce spectrum resources. By further optimizing the MBWC architecture (specialized meta-atoms for faster switching[35,37], multi-bit-programmable meta-atoms, a larger programmable metasurface,

**Table 2 Selected key parameters of the designed MBWC demo system based on a software-defined radio (Ettus USRP X310).**

| Parameter | Value |
|---|---|
| Wi-Fi working frequency | 2.442 GHz |
| Wi-Fi protocol | 802.11a/b/g |
| Wi-Fi router model | TL-WDR5620 |
| Modulation method | BPSK |
| Sampling rate | 40 Mbps |
| Transmission rate | 500 kbps |
| Frame size | 816,000 samples |
| PerFrame size | 320 × 320 pixel |
| Bit error rate | 0.005 |
| Transmit power | −10 dBm |
| Receiver gain | 20 dBm |

more receiver's on Bob's side, etc.) the information throughput of MBWC, and thereby the spectral efficiency, could be improved further.

As third and final proof-of-concept experiment, we demonstrate MBWC with quadrature PSK (QPSK) (de)modulation. The constellation results of MBWC-BPSK and MBWC-QPSK are compared in Fig. 5a. All four states are clearly distinguishable, the corresponding clouds in the complex plane do not overlap. Ref. [41] suggests that the similar results can also be achieved in more strongly reverberating indoor environments. The experimental results for a full-color image transfer with MBWC-BPSK and MBWC-QPSK are shown in Fig. 5b.

## Discussion

In summary, we introduced an MBWC concept in which the transmitter does not include a few dipole antennas but is based on a programmable metasurface, which modulates the propagation environment of ambient stray waves to encode digital information. We provided a theoretical framework for encoding/decoding and modulation/demodulation, and we built several proof-of-principle prototype systems tailored to the use of ambient commodity 2.4 GHz Wi-Fi signals. Antenna aperture and number of degrees of freedom of our programmable metasurface exceed those of traditional backscatter communication systems by three and a few hundred orders of magnitude, respectively. The resulting wave control enables focusing and multi-channel schemes, which allowed us to demonstrate unprecedented data rates on the order of hundreds of kbps and communication security—without requiring an active radiofrequency chain, energy and spectral resources to generate a carrier signal. Therefore, the reported MBWC strategy has the potential to fundamentally resolve pressing issues in conventional AWC (energy consumption, spectrum allotment, hardware cost, information security) and in conventional backscatter communication (low-data rates, lack of security), notably in the context of green IoT connectivity. We believe that our MBWC strategy provides a fundamentally new view on the role of programmable metasurfaces in wireless communication that can impact a wide range of future communication systems at radiofrequencies and beyond.

## Methods

**Programmable metasurface.** The designed 1-bit electronically controllable meta-atom is detailed in Supplementary Note 5. The meta-atom is composed of two substrate layers, and the top square patch is integrated with a SMP1345-079LF PIN diode. The top square patch size is $54 \times 54$ mm$^2$, the thickness of the top substrate F4BM is 1.5 mm along with the relative permittivity of 2.55 and the loss tangent of 0.0015 and the bottom substrate FR-4 has the thickness of 0.7 mm. Supplementary Fig. 2b, c report the experimental phase and amplitude

response of a unit cell as function of frequency for different unit cell configurations. The phase response changes by 180° around 2.4 GHz when the PIN is switched from OFF (ON) to ON (OFF), while the amplitudes remains almost unaltered (above 85%). The designed metasurface can to reshape the spatial distribution of ambient stray Wi-Fi waves.

**Information encoding/decoding and modulation/demodulation.** As detailed in Supplementary Note 1, the demodulated signal $H_{s \to meta \to mr}$ which conveys the digital information can be approximately expressed as:

$$H_{s \to meta \to mr}(\mathbf{r}_n, \omega, \mathcal{C}) \approx \sum_{n=1}^{N} \sigma(\mathbf{r}_n, \mathcal{C}) G(\mathbf{r}_n, \mathbf{r}_m, \omega), \qquad (5)$$

where $\sigma(\mathbf{r}_n, \mathcal{C})$ denotes the reflection coefficient of the meta-atom at $\mathbf{r}_n$ when the metasurface is controlled with the coding pattern $\mathcal{C}$, $G(\mathbf{r}_n, \mathbf{r}_m, \omega)$ is the Green's function imposed by the surrounding medium, $\mathbf{r}_m$ is the location of the master receiver, and $N$ is the total number of meta-atoms. $H_{s \to meta \to mr}(\mathbf{r}_m, \omega, \mathcal{C})$ approximately equals the signal radiated from the programmable metasurface with coding pattern $\mathcal{C}$. $H_{s \to meta \to mr}$ can be designed to have a finite number of well-distinguishable digital states by optimizing the coding patterns of the metasurface. These distinguishable digital states can be recognized by using a classifier or decoder. Moreover, Eq. (5) implies that $H_{s \to meta \to mr}$ can be designed to be radiation-direction-dependent, such that the information retrieved by different users can be independently controlled.

Focusing on the master receiver(s) is crucial to ensures a high SNR of the demodulated signal by harvesting the energy of stray Wi-Fi waves for MBWC. We have performed an additional analysis of channel strength and transmission efficiency, which is detailed in Supplementary Note 6. In the free-space far-field approximation, the smallest achievable size of the focal spot around the master receiver is on the order of $O(\lambda R/D)$ in the transverse direction, where $\lambda$ denotes the operational wavelength, $R$ is the communication distance, and $D$ is the size of the metasurface aperture. In strongly reverberant propagation environments the wavefield is speckle-like with a coherence length on the order of $\lambda/2$[24,46]. Consequently, in a reverberant medium the focal spot size may be much smaller than in its free-space counterpart. The smallest achievable focal spot size can be further reduced, irrespective of whether the propagation medium is free space or a reverberant environment, by placing the antenna inside a random distribution of scatterers as in ref. [47].

## Data availability
The data that support the findings of this study are available from the last author upon request.

## Code availability
Code that supports the findings of this study is available upon reasonable request from the last author.

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

## Acknowledgements

This work was supported by the National Key Research and Development Program of China under Grant nos. 2017YFA0700201, 2017YFA0700202, and 2017YFA0700203, and the 111 Project under Grant No. 111-2-05.

## Author contributions

L.L. conceived the idea, conducted the theoretical analysis, and wrote the paper. H.Z. built the proof-of-principle prototype system, Y.S. and H.Z. conducted experiments and data processing. L.L. and P.d.H. contributed to conceptualization and write-up of the project. All authors participated in the data analysis and read the paper.

## Competing interests

The authors declare no competing interests.
