## [Peer Review File · Nature Communications]

Reviewer #1 (Remarks to the Author):

In this paper, the authors propose a method to achieve passive wireless communication with a metamaterial panel by using opportunistic wi-fi signals. It is one new application of the smart walls or engineered tunable metamaterial panels that can be tuned to focus wireless signals onto a wireless device. The authors propose an original communication scheme based on the use of such panels. The idea could be of interest and could deserve publication, but I have a few fundamental questions to be addressed by the authors before I can make a recommendation.

1)

The minimal distance required between the two antennas (master receiver and slave receiver) is a critical problem. It should be large enough to allow for selective focusing on the master receiver. This problem is briefly addressed experimentally in Supplementary Note 6, in which the distance is more than one meter. From the practical point of view, this distance seems very large to be useful for typical objects and typical indoor applications.

From the theoretical point of view, it seems to be much larger than the focal spot radius given in the paper. Moreover: the formula of the focal spot radius $\lambda R/D$ is a transverse radius; the longitudinal radius is larger, of the order of $\lambda R^2/D^2$.

2)

The authors state (in several different ways):

"Our experiments demonstrate wireless communication without any active radio components".

However, the method requires to know with precision the positions of the receiver antennas. Therefore, the receiver should find a way to give its relative position with respect to the metamaterial panel with relative precision (at the scale of the focal spot). How is it supposed to do it without any active radio component ?

3)

The role of the type of the source is not sufficiently addressed in my point of view.

An analysis is given in the case of one point-like source in free space; a complementary analysis is given with a set of uniformly distributed random sources.

However, a typical indoor situation will have one or a few point-like sources, with multipathing. In this case some of the statements such as "an important conclusion that, wherever the non-cooperative wireless source is, its energy can be well focused toward the master receiver at r_m by choosing a suitable control coding pattern of the metasurface" are not clear at all.

4)

A few statements are given without justification:

- page 8: "The proposed PWC scheme is expected to virtually not affect the background AWC."

- supplementary page 8 : "For this case, the machine learning (ML) technique can be applied to find a ML agent for the Green's function."

5)

typo (S13): $r_A \rightarrow r_m$, $r_B \rightarrow r_S$

typo page S14: metasurace

I think that some early papers on the use of tunable metamaterial panels should be cited, such as:

N. Kaina, M. Dupré, G. Lerosey, M. Fink,

Shaping complex microwave fields in reverberating media with binary tunable metasurfaces, Scientific Reports, volume 4, 6693 (2015).

Reviewer #2 (Remarks to the Author):

The manuscript entitled "Metasurface-Assisted Passive Wireless Communication with Commodity Wi-Fi Signals" introduce a "passive wireless communication" (PWC) architecture, which is totally different from the traditional active wireless communication (AWC), by a coding metasurface. The concept presented in the manuscript really attracts me and I strongly believe such a SIMO (e.g. single-input multi-output) wireless communication technique will bring many enticing applications and might change our daily life in the era of IOT, 5G or even 6G communication in the future. Therefore, I believe the conceptual innovation of this paper meets the criteria of Nature Communications.

However, I still have following questions and doubt after I read the manuscript and supplementary materials. My questions are as following.

1. Through the manuscript, the key innovation should be the PWC architecture. As shown in Fig. 1c, the ambient stray wireless signals were utilized as the input power flow. However, the coding metasurface is still active indeed. I mean, in contrast to the modulator and LO used in the AWC, a FPGA-based encoder is used in the PWC. To control the coding process in the FPGA and switch "on"- "off" state of diode in the metasurface, power supply is still required in this so-called PWC architecture. Therefore, it seems the present architecture is still "active". I can understand that authors use the word "passive" because the input power of signal is from environment rather than their metasurface, but the "passive wireless communication" may introduce misunderstanding to some extent.

2. The statement "(ii) directly modulating the ambient stray electromagnetic waves with high signal-to-noise ratio (SNR)" is lack of evidence. Authors had better to explain it in detail. For example, the SNR might be influenced by following points: A) the gain of transmitting antenna at Alice side and B) the transmission power in the wireless channel. If we talk about point A, it is not the advantage of PWC. It is about the performance of RF hardware. If we change the dipole in Fig. 1b for AWC architecture to a beam-scanning antenna with large apertures, just like the metasurface used for PWC architecture, maybe the AWC will have a very high SNR as well. Regarding to point B, if the radiating power of wifi router in the PWC case is identical to that for the AWC case and they work in the same environment (e.g., they may share the same wireless channels and the similar noise), I am not sure whether the power of coded signal is high or low in the case of PWC compared to the traditional AWC case This is because that the coding metamaterial might receive only a few energy from the stray wireless signals. Is it possible to provide some additional result to prove this conclusion?

3. It seems that authors use equation (3) to demodulate PWC signals from unknown wireless signals. Is this for general cases? Is it just for single path or LOS (line-of-sight) case? If multi-path influence and NLOS (non-line-of-sight) should be considered when the environment is very complicate, should the PWC operational principle change? Under an NLOS case, will the signal be hard to distinguished due to the multiple reflected path?

4. Authors present very sufficient experimental results to support their PWC concept and show that the transmission rate of their PWC demo system is 500 Kbps. Is this transmission rate equivalent to the throughput (or similar concept) of a wireless communication system? According to my knowledge, throughput is one of the most important factors in the present wireless communication systems (base station and UEs). Therefore, is it possible to compare this transmission rate/throughput between AWC and PWC architectures with the use of the same wifi router? Of course, the wifi router in both cases should be arranged at the same position. Furthermore, what is the spectral width used in the experiment? How much is the spectrum efficiency in the experiment? Please compare it to the PWC case as well. If the throughput and spectrum efficiency could be higher than those in the AWC case, authors' PWC concept could attract much more interest from industrial engineers.

Reviewer #3 (Remarks to the Author):

The main problem of this manuscript is that the claim of the authors that they "introduce the fundamentally different concept of passive wireless communication (PWC)" is really too much since the papers published on this topic are very many. It appears that there are even start-up companies that develop this technique.

In fact, some earlier work was discussed in one of the papers in the list of references, ref. 27. Moreover, in the same reference it is mentioned that the principle is the same as spatial modulation (there is a lot of papers on that subject also) and backscattering.

Here are some references on papers that presented and developed this principle of passive wireless communications (the first one is a review, with many relevant references in its list, and the second one uses WiFi, the same approach as in this manuscript):

C. Xu, L. Yang, P. Zhang, Practical backscatter communication systems for battery-free Internet of Things: a tutorial and survey of recent research. *IEEE Sig. Process. Mag.* 35(5), 16–27 (2018)

J. Zhao, W. Gong, J. Liu, Spatial stream backscatter using commodity WiFi. *ACM MobiSys* (2018).
<https://dl.acm.org/citation.cfm?id=3210240.3210329>

There are research works on the same principle using WiFi, TV signals, 4G, and under research 5G signals. Here is an example of an early system that utilizes TV signals:

Vincent Liu, Aaron Parks, Vamsi Talla, Shyamnath Gollakota, David Wetherall, Joshua R. Smith, Ambient Backscatter: Wireless Communication Out of Thin Air, *SIGCOMM'13*, August 12–16, 2013, Hong Kong, China
<http://conferences.sigcomm.org/sigcomm/2013/papers/sigcomm/p39.pdf>

In my opinion, the authors need to be better position their research work.

I would like to note also that this work is similar to an earlier publication from same team: W. Tang, J. Y. Dai, M. Z. Chen, K.-K. Wong, X. Li, X. Zhao, S. Jin, Q. Cheng, and T. J. Cui, "MIMO transmission through reconfigurable intelligent surface: System design, analysis, and implementation", *IEEE J. Sel. Commun.*, 2020. [Online]. Available: <https://arxiv.org/pdf/1912.09955.pdf>.

There, the principle is similar, although the application is different. In that paper the authors use the surfaces as low complexity transmitters. In this submission, the surface is for backscattering applications. The function of the metasurface is similar, although not exactly the same.

As to the hardware realization, this varactor-switchable metasurface design was presented earlier by the same team in many publications, including papers in this journal (ref. 30).

On the positive note, this paper presents a working prototype whose construction is not an easy task, and the results may be of interest and, possibly, of some significance to the engineering community. However, in view of the lack of significant novelty, I would not recommend this manuscript for consideration by *Nature Communications*.

As an advice for paper revisions, I note that this paper is very difficult to read. There are lots of notations and abbreviations that are not generally known and used except perhaps some very specialized communities. For example, let's look at Fig. 1. What is "A-demodulator"? What is "P-demodulator"? What is "Antenna-S"? What is "PMS encoder", etc. It is very difficult to get an idea of the used approach from this picture.

Responses to Reviewers' Comments

We thank all reviewers for their constructive and valuable comments, which helped improve the quality of the manuscript significantly. We have revised the original manuscript according to the comments and suggestions, and all changes in the revised manuscript have been highlighted with yellow background. The replies to all comments are provided below in blue font.

To Reviewer #1:

Comment:

In this paper, the authors propose a method to achieve passive wireless communication with a metamaterial panel by using opportunistic wi-fi signals. It is one new application of the smart walls or engineered tunable metamaterial panels that can be tuned to focus wireless signals onto a wireless device. The authors propose an original communication scheme based on the use of such panels. The idea could be of interest and could deserve publication, but I have a few fundamental questions to be addressed by the authors before I can make a recommendation.

1)

The minimal distance required between the two antennas (master receiver and slave receiver) is a critical problem. It should be large enough to allow for selective focusing on the master receiver. This problem is briefly addressed experimentally in Supplementary Note 6, in which the distance is more than one meter.

From the practical point of view, this distance seems very large to be useful for typical objects and typical indoor applications.

From the theoretical point of view, it seems to be much larger than the focal spot radius given in the paper.

Moreover: the formula of the focal spot radius $\lambda R/D$ is a transverse radius; the longitudinal radius is larger, of the order of $\lambda R^2/D^2$.

Our Reply:

We thank the reviewer for this important comment. On the one hand, there are opportunities to significantly reduce the distance between two independent antennas by placing them inside a random distribution of scatterers. In essence, the presence of scatterers alters the effective index of medium such that two independent antennas may be much closer than the free-space diffraction limit. This idea was leveraged in Ref. 47 in combination with a time reversal scheme to achieve

deeply subwavelength focusing ($\lambda/30$). There is evidence from the optics community that similar subwavelength focusing can be achieved in combination with wave-front shaping, e.g. in Nature Photonics 4, 320–322 (2010). Our programmable metasurface may be interpreted to be analogous to a Spatial Light Modulator in optics. We strongly believe in the importance of the reviewer's question and the potential of the described solution, which is the subject of ongoing research.

On the other hand, we would like to point out that the focal spot size may be much smaller than the free-space focal spot if the propagation environment is highly reverberant (e.g. inside a vessel). Then, the environment is essentially a chaotic cavity and the wave field is speckle-like, with the typical speckle grain size being on the order of half wavelength. This comment shows that under certain conditions the minimal distance issue may be much less severe than that in free space.

We have expanded the corresponding paragraph in our manuscript accordingly:

Focusing on the master receiver(s) is crucial to ensure a high SNR of the demodulated signal by harvesting the energy of stray Wi-Fi waves for PWC. We have performed an additional analysis of channel strength and transmission efficiency which is detailed in **Supplementary Note 6**. In the **free-space** far-field approximation, the smallest achievable size of the focal spot around the master receiver is on the order of $O(\lambda R/D)$ **in the transverse direction**, where λ denotes the operational wavelength, R is the communication distance, and D is the size of the metasurface aperture. **In strongly reverberant propagation environments such as some indoor scenarios (e.g. inside a vessel, inside a room), the wave field is speckle-like with a coherence length on the order of $\lambda/2$.**^{24,46} **Consequently, in the reverberant medium the focal spot size may be much smaller than that in the free-space counterpart. The smallest achievable focal spot size can be further reduced, irrespective of whether the propagation medium is free space or a reverberant environment, by placing the antenna inside a random distribution of scatterers as in Ref. 47.**

Comment:

2)

The authors state (in several different ways):

"Our experiments demonstrate wireless communication without any active radio components".

However, the method requires to know with precision the positions of the receiver antennas. Therefore, the receiver should find a way to give its relative position with respect to the metamaterial panel with relative precision (at the scale of the focal spot). How is it supposed to do it without any active radio component ?

Our Reply:

We appreciate the reviewer's insightful remark. Our scheme relies on focusing ambient waves on the receiver. However, this does not inevitably mean that the receiver must actively communicate its position to the controller of the programmable metasurface.

One option to operate without direct feedback from the receiver is to leverage sensor fusion: the programmable metasurface may be equipped, for instance, with a Kinect-like camera that estimates the receiver's position in three-dimensional space. In the area of computational imaging, it is already customary to use optical sensors in combination with a meta-imager to identify the region of interest, see for instance Ref. 43.

Another option is to use techniques for sensing the position of non-cooperative objects. One such technique (see Ref. 27) leverages a programmable metasurface to ink unique wave fingerprints for distinct object positions with configurational diversity. Thus, the programmable metasurface could be used for position sensing in the first step before using it for PWC in the second step.

A further option to operate without direct feedback consists in leveraging nonlinear feedback from the receiver. Many RF chains inevitably include nonlinear components, otherwise it is possible to add one on purpose. Based on the nonlinear feedback, the metasurface can blindly focus ambient waves on the receiver – without even knowing the receiver's spatial location. This concept was demonstrated in Ref. 39.

There are hence several possibilities of using indirect feedback to focus on a receiver without the active radio component.

We have added the following paragraph in the revised manuscript:

It is important to note that the need for focusing on the receiver in our scheme does not inevitably mean that Bob must actively communicate his location to Alice. Instead, Alice can use a number of tricks to localize the position of non-cooperative Bob. A first option could leverage sensor fusion by equipping the metasurface with an optical camera with depth-sensing capability (e.g. Microsoft's Kinect) such that Alice can localize Bob. In computational microwave imaging, similar approach to the optical sensors to determine the region of interest has already become customary⁴³. A second option could leverage the programmable metasurface for a wave-fingerprint-based position sensing scheme as described in Ref. 27 before using the metasurface for wireless communication with our PWC proposal in the second step. A third option could rely on

nonlinear feedback from Bob's receiver: many radiofrequency chains inevitably include nonlinear components, otherwise it is also possible to add such a nonlinear element on purpose. As proposed in Ref. 39, based on such nonlinear feedback, Alice can blindly focus stray ambient waves on Bob's receiver – without even having estimated Bob's spatial position.

Comment:

3)

The role of the type of the source is not sufficiently addressed in my point of view.

An analysis is given in the case of one point-like source in free space; a complementary analysis is given with a set of uniformly distributed random sources.

However, a typical indoor situation will have one or a few point-like sources, with multipathing. In this case some of the statements such as "an important conclusion that, wherever the non-cooperative wireless source is, its energy can be well focused toward the master receiver at r_m by choosing a suitable control coding pattern of the metasurface" are not clear at all.

Our Reply:

The reviewer raises the important question of whether our technique can successfully be applied in propagation environments that do not resemble free space.

Our discussion in Supplementary Note 1 is very general and not restricted to the case that the propagation environment is free space. Indeed, the Green's function on which this analysis is based takes any multipath effects into account.

Alternatively, one may interpret the reflections as secondary sources such that a highly reverberant irregular environment is similar to a set of a very large number of uniformly distributed sources. We have shown in Supplementary Note 1 that for a very large number of uniformly distributed sources similar conclusions can be drawn as in the case of a single source.

Finally, we point out that from a practical point of view the focusing of waves in reverberant environments with programmable metasurfaces is possible and has been reported in several publications, including Refs. 24 and 39.

Nonetheless, the reviewer is correct that multipath scenarios are more challenging for our PWC scheme because we have no analytical or numerical expression for the Green's function that we would need to identify the optimized metasurface configurations. However, it is possible to train

an artificial neural network to approximate such a complicated Green's function – an approach that we already took in a different context in Ref. 31. Once such a learned forward model is available, we can simply replace the free space Green's function by the learned Green's function and identify the optimal coding patterns as before. The concept of the learned Green's function is elaborated in Supplementary Note 3 (see also our response to question 4b).

We have edited Supplementary Note 1 for enhanced clarity and added a few remarks specifically to point out the generality of the Green's function formalism:

The Green's function formalism is very general and accounts for any type of propagation environment. For instance, multipath effects as found in complex environments with scattering effects and reverberation, are accounted for.

...

These results, due to the generality of the Green's function formalism, are not restricted to a free space propagation environment. They are equally valid in propagation environments with multipath effects. Indeed, focusing a speckle-like ambient field with a programmable metasurface is also possible in reverberant environments, as demonstrated in Ref.¹.

Comment:

4)

A few statements are given without justification:

- page 8: "The proposed PWC scheme is expected to virtually not affect the background AWC."

Our Reply:

We thank the reviewer for alerting us to the need to reference and justify this statement. In order to strongly impact the background AWC, the signal modulated by the programmable metasurface would have to be very strong at the AWC access point. Yet, the metasurface-shaped signal is only strong in the vicinity of Bob's master receiver. Assuming that the AWC access point is not within the focal spot around Bob, it is therefore reasonable to assume that the effect of the programmable metasurface on the signal received by the AWC access point is negligible.

This argument is backed up by experimental evidence from the literature on ambient backscatter communication: the normal Wi-Fi network performance is negligibly affected unless the backscatter device is extremely close to the Wi-Fi access point in which case the network throughput is reduced by a few percent. See, for instance, Ref. 10.

We have added the following justification to our manuscript:

In order to affect the background AWC, the signal modulated by the programmable metasurface would have to be very strong at the AWC access point which is not the case unless the AWC access point is in close vicinity of Bob's master receiver. Similar conclusions on the negligible impact on the background AWC have been reached in the literature on the traditional ambient backscatter communication¹⁰.

Comment:

- supplementary page 8 : "For this case, the machine learning (ML) technique can be applied to find a ML agent for the Green's function."

Our Reply:

We thank the reviewer for alerting us to the fact that we forgot to reference this statement. We demonstrated in the previous work (Ref. 31) that in propagation environments that cannot be approximated as free space, it is possible to train an artificial neural network to approximate the Green's function.

The revised sentence reads as follows:

Explicitly generalizing the above discussion to a complex propagation environment is very challenging since typically no tractable models are available for such scenarios, i.e. no analytical or numerical expression for the Green's function is available. Nonetheless, it is possible to leverage machine-learning tools to train an artificial neural network to approximate the Green's function, as in Ref.².

Comment:

5)
typo (S13): r_A->r_m, r_B->r_S
typo page S14: metasurface

Our Reply:

We thank the referee for spotting these typos. We have corrected these typos.

Comment:

I think that some early papers on the use of tunable metamaterial panels should be cited, such as: N. Kaina, M. Dupré, G. Lerosey, M. Fink, Shaping complex microwave fields in reverberating media with binary tunable metasurfaces, Scientific Reports, volume 4, 6693 (2015).

Our Reply:

We agree that it may be helpful for the reader to be made aware of earlier work involving programmable metasurfaces. The cited paper is included in the reference list in the revised manuscript.

To Reviewer #2

Comment:

The manuscript entitled “Metasurface-Assisted Passive Wireless Communication with Commodity Wi-Fi Signals” introduce a “passive wireless communication” (PWC) architecture, which is totally different from the traditional active wireless communication (AWC), by a coding metasurface. The concept presented in the manuscript really attracts me and I strongly believe such a SIMO (e.g. single-input multi-output) wireless communication technique will bring many enticing applications and might change our daily life in the era of IOT, 5G or even 6G communication in the future. Therefore, I believe the conceptual innovation of this paper meets the criteria of Nature Communications.

However, I still have following questions and doubt after I read the manuscript and supplementary materials. My questions are as following.

1. Through the manuscript, the key innovation should be the PWC architecture. As shown in Fig. 1c, the ambient stray wireless signals were utilized as the input power flow. However, the coding metasurface is still active indeed. I mean, in contrast to the modulator and LO used in the AWC, a FPGA-based encoder is used in the PWC. To control the coding process in the FPGA and switch “on”-“off” state of diode in the metasurface, power supply is still required in this so-called PWC architecture. Therefore, it seems the present architecture is still “active”. I can understand that authors use the word “passive” because the input power of signal is from environment rather than their metasurface, but the “passive wireless communication” may introduce misunderstanding to some extent.

Our Reply:

We thank the reviewer for raising an important question regarding the notion of “passive”. First, we would like to point out that the power needed to control the metasurface is minimal, see for instance the estimates in Ref. 24. As the reviewer points out, we use the terms “active” and “passive” with respect to the generation of electromagnetic waves rather than with respect to the reflection properties of the programmable metasurface. If “passive” was taken to mean that all components of the scheme (including the metasurface) are static (i.e. not reconfigurable), it would be impossible to achieve any communication by definition.

We have added the following clarifications to our manuscript:

The term “passive” refers to the origin of the carrier signals and does not mean that all components involved in the wireless infrastructure are static – otherwise by definition no information could be encoded.

...

The power needed to program the metasurface is minimal and can be as low as a few μW per meta-atom²⁴.

Comment:

2. The statement “(ii) directly modulating the ambient stray electromagnetic waves with high signal-to-noise ratio (SNR)” is lack of evidence. Authors had better to explain it in detail. For example, the SNR might be influenced by following points: A) the gain of transmitting antenna at Alice side and B) the transmission power in the wireless channel. If we talk about point A, it is not the advantage of PWC. It is about the performance of RF hardware. If we change the dipole in Fig. 1b for AWC architecture to a beam-scanning antenna with large apertures, just like the metasurface used for PWC architecture, maybe the AWC will have a very high SNR as well. Regarding to point B, if the radiating power of wifi router in the PWC case is identical to that for the AWC case and they work in the same environment (e.g., they may share the same wireless channels and the similar noise), I am not sure whether the power of coded signal is high or low in the case of PWC compared to the traditional AWC case This is because that the coding metamaterial might receive only a few energy from the stray wireless signals. Is it possible to provide some additional result to prove this conclusion?

Our Reply:

We appreciate that the reviewer alerts us to the need to clarify the cited statement. The short statement in our original manuscript was misleading since we do not intend to claim that the SNR of our PWC scheme is better than that of AWC. Instead, the cited sentence seeks to convey that within possible usages of the programmable metasurface for PWC, our technique to focus the field on the master receiver enables a comparatively high SNR. To clarify this point, in principle an alternative PWC scheme could consist in using two random metasurface configurations as \mathcal{C}_0 and \mathcal{C}_1 . However, such a choice would fail to achieve a high distinguishability of the two states in the (inevitable) presence of noise – see Fig. 1(d). Our technique ensures high distinguishability of the two states which guarantees a comparatively high SNR – in comparison to other potential PWC schemes.

We have added the following clarification to our manuscript:

The notion of distinguishability is intimately related to SNR of the PWC scheme: the more distinguishable the two coding patterns are, the higher SNR is on Bob's side. The coding patterns identified by solving the constrained optimization problem ensure a high SNR for PWC.

Comment:

3. It seems that authors use equation (3) to demodulate PWC signals from unknown wireless signals. Is this for general cases? Is it just for single path or LOS (line-of-sight) case? If multi-path influence and NLOS (non-line-of-sight) should be considered when the environment is very complicate, should the PWC operational principle change? Under an NLOS case, will the signal be hard to distinguished due to the multiple reflected path?

Our Reply:

We thank the reviewer for this interesting question. The Green's function formalism we used in Supplementary Note 1 to derive Eq. 3 of the main text is very general and applicable to any types of propagation environments, either in free space or in a multipath scenario. Therefore, the PWC demodulation scheme described by Eq. 3 remains valid in NLOS scenarios. Nonetheless, the reviewer is correct that NLOS scenarios are more challenging for our PWC scheme because we have no analytical or numerical expression for the Green's function that we would need to identify the optimized metasurface configurations. However, it is possible to train an artificial neural network to approximate such a complicated Green's function – an approach that we already took in a different context in Ref. 31. Once such a learned forward model is available, we can simply replace the free space Green's function by the learned Green's function and identify the optimal coding patterns as before.

We have edited Supplementary Note 1 for enhanced clarity and to explicitly point out the generality of the Green's function formalism:

The Green's function formalism is very general and accounts for any type of propagation environment. For instance, multipath effects as found in complex environments with scattering effects and reverberation, are accounted for.

...

These results, due to the generality of the Green's function formalism, are not restricted to a free space propagation environment. They are equally valid in propagation environments with multipath effects. Indeed, focusing a speckle-like ambient field with a programmable metasurface is also possible in reverberant environments, as demonstrated in Ref. 1.

We have also edited Supplementary Note 3 to convey the concept of a learned Green's function more clearly:

Explicitly generalizing the above discussion to a complex propagation environment is very challenging since typically no tractable models are available for such scenarios, i.e. no analytical or numerical expression for the Green's function is available. Nonetheless, it is possible to leverage machine-learning tools to train an artificial neural network to approximate the Green's function, as in Ref. 2.

Comment:

4. Authors present very sufficient experimental results to support their PWC concept and show that the transmission rate of their PWC demo system is 500 Kbps. Is this transmission rate equivalent to the throughput (or similar concept) of a wireless communication system? According to my knowledge, throughput is one of the most important factors in the present wireless communication systems (base station and UEs). Therefore, is it possible to compare this transmission rate/throughput between AWC and PWC architectures with the use of the same wifi router? Of course, the wifi router in both cases should be arranged at the same position. Furthermore, what is the spectral width used in the experiment? How much is the spectrum efficiency in the experiment? Please compare it to the PWC case as well. If the throughput and spectrum efficiency could be higher than those in the AWC case, authors' PWC concept could attract much more interest from industrial engineers.

Our Reply:

We thank the reviewer for raising an important question about spectrum efficiency. Indeed, one important problem of our modern society is spectrum allotment: we cannot use more and more bandwidth to meet the ever-increasing demand for more information throughput with conventional AWC. Here, PWC offers a solution: without using any additional spectral resources (only those used for AWC anyway), significant additional throughputs of information can be achieved, foreseeably on the same order as the information throughput of AWC.

The transmission rate of our proof-of-concept demo PWC system (Supplementary Note 8) is almost comparable to that of a standard AWC system. In our demo system, a Wi-Fi router (Candy) sends information to a conventional Wi-Fi user (Davy). The waves used for this traditional AWC information transfer are used to additionally transfer information from Alice to Bob. That is, the waves received by Bob were generated by Candy but modulated by Alice. The Wi-Fi router

considered in our experiments uses frequencies ranging from 2.432GHz to 2.452GHz, i.e., it operates with a spectral bandwidth of 20MHz, and it has a transmission rate of 2Mbps. The transmission rate of BPSK-based PWC is 500Kbps in our demo system.

We note that our PWC demo has not been heavily optimized; indeed, there are multiple opportunities to enhance the transmission rate of PWC. First, a specialized programmable metasurface can be designed whose meta-atoms achieve a higher switching speed. In this way, the pulse response of the meta-atom can be designed in order to reduce the inter code interference to achieve a higher PWC transmission rate. Furthermore, the meta-atom programmability could be increased from 1 bit (i.e. only two possible states) to multi-bit²⁸. Second, more PWC receiving antennas can be deployed, possibly in combination with a larger programmable metasurface, to enhance the transmission rate.

The throughput of our PWC prototype is thus already close to that of modern AWC and could be further enhanced, as outlined above.

We believe that spectrum efficiency should not be evaluated separately for PWC: instead, the spectrum efficiency of AWC alone should be compared with the spectrum efficiency of AWC and PWC together. Since PWC makes use of the same spectral resources that are already used for AWC anyway, any information transfer by PWC improves spectral efficiency. In our prototype, we improved spectral efficiency by 25% since on top of AWC's 2Mbps we transferred additional 500Kbps by PWC – without using any additional spectral resources. With further optimization, we faithfully expected that PWC can at least double the spectral efficiency.

We added the following clarification to our manuscript:

It is important to note that PWC achieves this information transfer without using any additional spectral resources since it only makes use of already existing waves from AWC. The use of PWC has therefore the potential to help our modern society to make more efficient use of scarce spectrum resources. By further optimizing the PWC architecture (specialized meta-atoms for faster switching^{35,37}, multi-bit-programmable meta-atoms²⁸, a larger programmable metasurface, more receiver's on Bob's side, ...) the information throughput of PWC, and thereby the spectral efficiency, could be improved further.

To Reviewer #3

Comment:

The main problem of this manuscript is that the claim of the authors that they "introduce the fundamentally different concept of passive wireless communication (PWC)" is really too much since the papers published on this topic are very many. It appears that there are even start-up companies that develop this technique.

In fact, some earlier work was discussed in one of the papers in the list of references, ref. 27. Moreover, in the same reference it is mentioned that the principle is the same as spatial modulation (there is a lot of papers on that subject also) and backscattering.

Here are some references on papers that presented and developed this principle of passive wireless communications (the first one is a review, with many relevant references in its list, and the second one uses WiFi, the same approach as in this manuscript):

C. Xu, L. Yang, P. Zhang, Practical backscatter communication systems for battery-free Internet of Things: a tutorial and survey of recent research. *IEEE Sig. Process. Mag.* 35(5), 16–27 (2018)

J. Zhao, W. Gong, J. Liu, Spatial stream backscatter using commodity WiFi. *ACM MobiSys* (2018). <https://dl.acm.org/citation.cfm?id=3210240.3210329>

There are research works on the same principle using WiFi, TV signals, 4G, and under research 5G signals.

Here is an example of an early system that utilizes TV signals:

Vincent Liu, Aaron Parks, Vamsi Talla, Shyamnath Gollakota, David Wetherall, Joshua R. Smith, Ambient Backscatter: Wireless Communication Out of Thin Air, *SIGCOMM'13*, August 12–16, 2013, Hong Kong, China

<http://conferences.sigcomm.org/sigcomm/2013/papers/sigcomm/p39.pdf>

In my opinion, the authors need to be better position their research work.

Our Reply:

We thank the reviewer for bringing these very interesting references to our attention. We fully agree that our work must be clearly positioned relative to the existing work. Backscatter

communication is of course a well-established field, dating back at least to the famous Great Seal bug in the 1940s and is the backbone of the current RFID technologies. We agree that such backscatter communication schemes should be mentioned and classified as “passive”, too.

Nonetheless, there are important conceptual differences between these schemes and our proposal.

First, our work is the first to combine the backscatter communication with the programmable metasurface concept. The conventional backscatter schemes leverage an antenna to capture a wave and to re-emit a modulated wave. See, for instance, Fig. 3 in Ref. 12. In contrast, our “backscatter” **PWC transmitter does not include an antenna**. It relies on modulating the propagation environment with the programmable metasurface, giving us much greater control over the backscattered signal.

Second, it is the programmable metasurface that enables us to **exclusively communicate with designated targets**. In our scheme, the information encoded by Alice can only be received within the focal spot where Bob is located. An eavesdropper outside the focal spot would not be able to receive any information. In fact, our scheme would allow us to **deliberately send misleading information to an eavesdropper at the same time** as the useful information is sent to Bob. In contrast, the conventional backscatter schemes do not target their signal to a specific location in space and are hence not secure.

Third, owing to the much larger controls over the backscattered field (essentially, our aperture is huge compared to the traditional backscatter antenna), our metasurface enables much **higher bit rates** than the conventional backscatter techniques. Our proof-of-principle experiment without much optimization already yielded 500 kbps. Most of the cited works from the backscatter literature only operate between 100 bps and 10 kbps.

While there are certainly some startup companies interested in metasurfaces as well as startup companies interested in backscatter communications, none of these companies (to the best of our knowledge) works on the idea put forward in this manuscript. More importantly, none of these companies has published anything similar to the idea proposed in this manuscript.

We have added the following paragraphs to the introduction of our manuscript to clearly position our work with respect to the conventional backscatter communication:

To fundamentally address these challenges, here we take inspiration from backscatter communication system⁹⁻¹². The underlying idea is well known at least since the infamous Great Seal

Bug¹³ and has become the backbone of modern radiofrequency identification (RFID): an antenna captures a deliberately emitted or opportunistic ambient signal and the information can be encoded in the reflected and re-emitted signals by modulating the antenna impedance. Such a system, especially when it leverages already existing stray ambient waves (e.g. Wi-Fi, TV signals, and 4G), may be characterized as *passive wireless communication* (PWC) since no active carrier signal is launched. The limited antenna aperture, inherent in the use of a single or few antennas, results in severe restrictions on the achievable data rates and wave control. Nonetheless, these techniques hold great promise for low-data-rate communication for RFID and IoT sensors. Moreover, from the standpoint of communication security, one notes that these techniques are unable to restrict their communication to designated target receivers, let alone to send deceiving information to an eavesdropper.

Here, we propose a PWC scheme in which the transmitter does not include an antenna but relies on modulating the propagation environment with a programmable metasurface^{14,15}: digital information is encoded into omnipresent stray ambient waves on the physical level using the programmable metasurface in an inexpensive and dynamic manner. In contrast to the existing backscatter communication schemes, our proposal offers a significantly larger control over the waves, enabling much higher data rates and secure communications.

We have also modified the abstract accordingly:

Here, we introduce a fundamentally different concept for *passive* wireless communication (PWC) which has the potential to resolve the above-mentioned problems. Unlike the traditional backscatter communication systems that modulate the antenna's impedance, the proposed PWC transfers the digital information in high security and at high speed by modulating/demodulating the propagation environment of already existing ambient stray electromagnetic waves using a programmable metasurface.

We also modified the first sentence of our conclusion to specify the distinguishing nature of our PWC technique:

In summary, we introduced a PWC concept in which the transmitter does not include an antenna but is based on a programmable metasurface which modulates the propagation environment of ambient stray waves to encode digital information.

Comment:

I would like to note also that this work is similar to an earlier publication from same team: W. Tang, J. Y. Dai, M. Z. Chen, K.-K. Wong, X. Li, X. Zhao, S. Jin, Q. Cheng, and T. J. Cui, "MIMO transmission through reconfigurable intelligent surface: System design, analysis, and implementation", IEEE J. Sel. Commun., 2020. [Online]. Available: <https://arxiv.org/pdf/1912.09955.pdf>.

There, the principle is similar, although the application is different. In that paper the authors use the surfaces as low complexity transmitters. In this submission, the surface is for backscattering applications. The function of the metasurface is similar, although not exactly the same.

Our Reply:

We agree with the reviewer that the cited paper and our manuscript both consider the use of programmable metasurfaces in a wireless communication context. However, they have two major differences.

First, the cited work belongs to the *active* wireless communication, where the *active* source of electromagnetic waves is an essential component of the transmitter setup. Note that in the cited work, a feeding-horn antenna was carefully aligned to ensure that a plane wave is normally incident on the metasurface. Essentially, one has to interpret the metasurface plus the feed antenna as the overall transmitter, where the metasurface configuration as well as the feed antenna type, orientation, and position are jointly optimized. In contrast, our work, as stated by the reviewer, is concerned with a backscatter context, in which no knowledge about the source is available. We do not need to control the type, orientation, and position of the source.

Second, in the cited work the programmable metasurface was based on time-domain digital coding, and the information was encoded in the time domain. However, in our present manuscript the programmable metasurface is based on time-space-domain digital coding, and the information is encoded in the time-space codes. The time-domain coding metasurface is used to control the spectra (harmonics) of the electromagnetic waves, while the time-space-domain coding metasurface is used to control jointly both the EM spatial beams and the temporal information with high security.

We have added the following comment to our revised manuscript:

Conceptually, the use of the metasurface to program the propagation environment of waves with unknown characteristics (e.g. source location, angle of arrival, shape of incident wave front)

sharply differs from the existing communication scheme⁴³, where a programmable metasurface and a feeding-antenna source must be seen as whole of the transmitter, rather than the propagation environment.

Comment:

As to the hardware realization, this varactor-switchable metasurface design was presented earlier by the same team in many publications, including papers in this journal (ref. 30).

Our Reply:

It is correct that we have previously used similar programmable metasurfaces in other contexts such as intelligent gesture recognition. However, the focus of the present work is not on the design of the programmable metasurface itself nor on the previously considered applications. The programmable metasurface is a tool that we leverage to demonstrate a new idea for passive wireless communication. This is also the great merit of the programmable metasurface: the same or similar hardware can produce numerously different functionalities.

Comment:

On the positive note, this paper presents a working prototype whose construction is not an easy task, and the results may be of interest and, possibly, of some significance to the engineering community. However, in view of the lack of significant novelty, I would not recommend this manuscript for consideration by Nature Communications.

Our Reply:

We thank the reviewer for appreciating the technical effort that went into our proof-of-concept experiments.

The reviewer's comments helped us to position our work more carefully with respect to the existing literature on backscatter communications. We hope that our replies and revision of the manuscript have clearly clarified the novelty of our technique, and the aspects that our scheme sharply differs from the existing literatures.

Comment:

As an advice for paper revisions, I note that this paper is very difficult to read. There are lots of notations and abbreviations that are not generally known and used except perhaps some very specialized communities. For example, let's look at Fig. 1. What is "A-demodulator"? What is "P-

demodulator”? What is “Antenna-S”? What is “PMS encoder”, etc. It is very difficult to get an idea of the used approach from this picture.

Our Reply:

Sorry for these confusions.

We appreciate the reviewer’s advice and we have revised Fig. 1 accordingly.

Reviewer #1 (Remarks to the Author):

I find that the authors have satisfactorily taken into account my remarks. I am happy to recommend the paper for publication.

Reviewer #2 (Remarks to the Author):

I am satisfied with authors' replies to my comment 2 and 3. However, I still think the replies to my comment 1 and 4 may cause some misleading.

In comment 1, I questioned the definition of "passive". The authors give a definition fitting current work and introduce the application scenario of the given definition. However, the replies regarding the issue are not entirely convincing. What is the physical meaning of a passive system? Indeed, comment 1 from reviewer #1 also queries the same issue. Maybe it would be better with a reasonable explanation through rigorous physical model. It could be literatures or others. According to the authors' answer, it seems more accurate to define the device as low-power consumption rather than as passive system. I do not think it is a "static" system, because there are FPGA parts inside. This is the key related to the most significant conceptual novelty in the manuscript. I suggest the authors elaborate on this issue more carefully.

In comment 4, I questioned the performance comparison with previous/conventional AWC architecture. According to the data provided in the revision, it seems that the performance of a PWC architecture along is worse than that of the AWC architecture at present stage. If AWC and PWC can be overlapped/integrated in the same physical channel perfectly, author expected "doubled spectral efficiency". Do this claim follow Shannon-Hartley theorem? Furthermore, will there be correlation or mutual coupling between the codes transferred through two architectures?

Reviewer #3 (Remarks to the Author):

in the revised version, the authors write that the difference with respect to earlier works on such systems is that instead of a modulated "antenna" they use a "programmable metasurface": "we propose a PWC scheme in which the transmitter does not include an antenna but relies on modulating the propagation environment with a programmable metasurface".

But this "programmable metasurface" is, in fact, the same as tunable reflectarray ANTENNA. There is no difference between "programmable" and "modulated" or "tunable" or between "reflectarray" and "metasurface" (at least for this application). Modulating any passive reflector (using the wording in earlier works), one "modulates the propagation environment" (as it is now called in this paper).

Depending on thought application, modulated and tunable reflector antennas can be weakly directed (like a dipole, as in the earliest works) or large-aperture, highly directed reflectarrays, as in this version.

Unfortunately, I still do not see any fundamental, conceptual novelty in this paper, and my opinion does not change.

Response Letter

Reviewer #1:

I find that the authors have satisfactorily taken into account my remarks. I am happy to recommend the paper for publication.

Response:

We are delighted to read that Reviewer #1 approves the publication of our manuscript. Thank you very much.

Reviewer #2:

I am satisfied with authors' replies to my comment 2 and 3. However, I still think the replies to my comment 1 and 4 may cause some misleading.

In comment 1, I questioned the definition of “passive”. The authors give a definition fitting current work and introduce the application scenario of the given definition. However, the replies regarding the issue are not entirely convincing. What is the physical meaning of a passive system? Indeed, comment 1 from reviewer #1 also queries the same issue. Maybe it would be better with a reasonable explanation through rigorous physical model. It could be literatures or others. According to the authors' answer, it seems more accurate to define the device as low-power consumption rather than as passive system. I do not think it is a “static” system, because there are FPGA parts inside. This is the key related to the most significant conceptual novelty in the manuscript. I suggest the authors elaborate on this issue more carefully.

Response:

We understand the reviewer's continued concern with the term “passive”. To address the concern, we have replaced “passive wireless communication” (PWC) by “massive backscatter wireless communication” (MBWC) in the revised manuscript. We believe that this new term is more precise and avoids the confusion about what parts of our system are meant to be passive.

Please note that we use the new terminology in our responses to the remaining questions.

Additionally, the term “passive” only appears in the following sentence in **Results** section of the revised manuscript, avoiding any misleading and confusion:

In terms of the origin of the carrier signals, our MBWC technique is “passive”. However, the programmable metasurface configuration is of course not static – otherwise by definition no information could be encoded.

In comment 4, I questioned the performance comparison with previous/conventional AWC architecture. According to the data provided in the revision, it seems that the performance of a PWC architecture along is worse than that of the AWC architecture at present stage. If AWC and PWC can be overlapped/integrated in the same physical channel perfectly, author expected

“doubled spectral efficiency”. Do this claim follow Shannon-Hartley theorem? Furthermore, will there be correlation or mutual coupling between the codes transferred through two architectures?

Response:

We thank the reviewer for the insightful question on how our claim of doubling the spectral efficiency can be reconciled with the Shannon-Hartley theorem.

In its original version, the Shannon-Hartley theorem defines the maximum rate of information transfer (capacity) in a communication channel as $C = \log_2(1 + \rho)$ per frequency, where ρ is the signal-to-noise ratio (SNR) of the channel. Later on, researchers sought ways for more information transfer in spite of the limited spectral resources by using multiple channels (e.g., multiple OAM modes, multiple spatial channels, ...), which leads to the generalized version of the Shannon-Hartley theorem: $C = \log_2(\det(I + \frac{\rho}{n} HH^\dagger))$ per frequency, where $H \in \mathbb{C}^{M \times N}$ is the channel matrix. Based on a singular value decomposition of H , the generalized Shannon-Hartley theorem can be rewritten as $C = \sum_{i=1}^K \log_2(1 + \frac{\rho}{n} \sigma_i)$ per frequency, where K is the smaller one of the two dimensions of H . Note that the effective number of channels depends on the correlations between channels. In short, by using multiple channels, more information can be transferred within the same frequency band. That is to say, the spectral efficiency is higher without violating the Shannon-Hartley theorem.

In light of the generalized Shannon-Hartley theorem, our MBWC proposal can be interpreted as creating additional communication channels which link Alice’s programmable metasurface to Bob’s receiving antennas. These additional MBWC channels are used at the same frequencies as those of the background AWC. Thereby, more information can be transferred despite using the same frequency band. In this regard, our claim of potentially doubling the spectral efficiency by using MBWC on top of AWC does not violate the generalized Shannon-Hartley theorem.

The reviewer asks about the correlations that may exist between our MBWC channels and the regular AWC channel. The conception of our MBWC scheme implies that there is negligible mutual coupling between MBWC and AWC channels due to the following effects:

- (i) Separation of time scales. As detailed in the manuscript (see Equations 1-3 in the Main Text and Supplementary Notes 1 and 2), the information in MBWC is encoded on a slow-time scale τ whereas the information in AWC is encoded on a fast-time scale t .

The information of the background AWC is received exactly by the master antenna in our MBWC scheme such that Bob could in principle access the AWC information, which serves as the “carrier” signal for our MBWC. However, Bob’s demodulation procedure based on at least two antennas (master and slave antennas) throws away fast fluctuations, such that no AWC information is contained in his demodulated MBWC signal.

- (ii) The MBWC focal spot (i.e., the area within which MBWC information can be received and decoded) is limited to the Bob’s vicinity, as described in Equations 1-3 in our manuscript. The AWC user away from Bob cannot receive the MBWC information since he/she does not access to the focal spot.

In addition, the MBWC channels are mutually orthogonal by construction due to the unique capability of the programmable metasurface for flexible wavefront manipulation: focusing on Bob-R does not yield any undesired focusing on Bob-G or Bob-B, and so on. Therefore, we expect that the overall channel matrix accounting for both AWC and MBWC is roughly diagonal (i.e. no significant channel correlations).

We added the following clarification to the revised manuscript:

The MBWC demodulation procedure based on the separation of time scales therefore guarantees that no AWC information (fast time t) is contained in the decoded MBWC signal (slow time τ) – even though the information could in principle be accessed by the MBWC receiving antennas. We previously established that no MBWC information is contained in the received AWC signal since the AWC receiver is outside the focal spot at which Bob’s receiver is placed. Therefore, no significant coupling occurs between the information transferred with AWC and MBWC.

It is important to note that MBWC achieves this information transfer without using any additional spectral resources since it creates additional communication channels that make use of already existing waves from AWC.

Reviewer #3 (Remarks to the Author):

in the revised version, the authors write that the difference with respect to earlier works on such systems is that instead of a modulated "antenna" they use a "programmable metasurface": "we propose a PWC scheme in which the transmitter does not include an antenna but relies on modulating the propagation environment with a programmable metasurface".

But this "programmable metasurface" is, in fact, the same as tunable reflectarray ANTENNA. There is no difference between "programmable" and "modulated" or "tunable" or between "reflectarray" and "metasurface" (at least for this application). Modulating any passive reflector (using the wording in earlier works), one "modulates the propagation environment" (as it is now called in this paper). Depending on thought application, modulated and tunable reflector antennas can be weakly directed (like a dipole, as in the earliest works) or large-aperture, highly directed reflectarrays, as in this version.

Unfortunately, I still do not see any fundamental, conceptual novelty in this paper, and my opinion does not change.

Response:

We thank the reviewer for detailing the continued concern about how our work is presented in relation to the backscatter communication literature.

First, we would like to point out that the usage of the programmable metasurface in our system differs sharply from the conventional usage of a tunable reflectarray. The reflectarrays are usually used as part of a transmit architecture, i.e., they are carefully positioned relative to a feed antenna (e.g. a horn antenna) to shape the transmitted wave without needing multiple radiofrequency chains as in an antenna array. In our work, the programmable metasurface does **not** need to be carefully positioned relative to a feed antenna. In fact, our metasurface is programmed without knowing the source location or type of source (dipole, horn, etc.). Our scheme even works for multiple unknown sources, as detailed in Supplementary Note 1. In our scheme, the programmable metasurface is not a part of the transmit architecture but a part of the propagation environment. Therefore, while we understand that the reviewer believes that one could use the tunable reflectarray instead of our programmable metasurface to implement our proposed scheme, we would like to stress that the

usage of tunable reflectarray would not be equivalent to the well-known conventional reflectarray applications.

Second, in order to more clearly relate our work to the backscatter literature, we have replaced “passive wireless communication” (PWC) by “massive backscatter wireless communication” (MBWC) in the revised manuscript. The new term “massive” highlights what sharply distinguishes our work (using the programmable metasurface) from the existing backscatter schemes (using one or a few dipole antennas). In contrast to the existing backscatter communication schemes with a few degrees of freedom and an antenna aperture of a few cm^2 , our proposal has 2^{768} degrees of freedom (768 is the number of meta-atoms), and the aperture size is 2.2 m^2 . The drastically improved wave control in our work enables focusing and multi-channel schemes, which allows us to demonstrate an **unprecedented communication speed** on the order of hundreds of Kbps as well as to introduce **communication security** in backscatter communication – without requiring active radiofrequency chain, energy and spectral resources to generate the carrier signal. Therefore, the reported MBWC strategy has the potential to fundamentally resolve the pressing issues in the conventional AWC (energy consumption, spectrum allotment, hardware cost, information security) and in the conventional backscatter communication (low data rates, lack of security).

Third, besides the significant impact of our work on the backscatter communication community, our work is also of significant interests to the rapidly growing community of “reconfigurable intelligent surfaces” (RIS), in which our programmable metasurface is an example (see Ref. 33). The RIS community so far only considers the role of RIS to be related to changing the propagation environment of AWC. Our work opens up an entirely new perspective regarding the role that RIS may play in wireless communications by using RIS in a backscatter communication context to directly transmit information with the RIS itself.

In the abstract we made the following changes:

Backscatter communication systems, on the other hand, modulate an antenna’s impedance to encode information into the already existing waves but suffer from low data rates and lack of information security. Here, we introduce the concept of massive backscatter wireless communication (MBWC) which modulates the propagation environment of ambient stray electromagnetic waves using a programmable metasurface. The large aperture and huge number of degrees of freedom of the metasurface enable unprecedented wave control in the backscatter communication and thereby the secure and high-speed transfer of digital information.

...

At the same time, the concept is applicable to all types of wave phenomena and provides a fundamentally new perspective on **the role of programmable metasurfaces in** future wireless communication architectures.

In the main text we made the following changes:

Nonetheless, inherent in the use of a single or few antennas is a very limited antenna aperture and number of degrees of freedom, resulting in severe restrictions on the achievable wave control. The limited wave control prevents focusing and efficient multi-channel communication, a prerequisite for transferring large amounts of information within a restricted frequency band at high speed. Moreover, from the standpoint of communication security, the limited wave control implies an inability to restrict the communication to designated target receivers, let alone to send deceiving information to an eavesdropper.

...

In contrast to existing backscatter communication schemes with a few degrees of freedom and an antenna aperture of a few cm^2 , our proposal offers a significantly larger control over the waves. We use a 2.2 m^2 programmable metasurface consisting of 768 meta-atoms, such that our antenna aperture is larger by three orders of magnitude and our number of degrees of freedom is larger by several hundred orders of magnitude. The resulting unprecedented wave control enables much higher data rates and the implementation of secure communication protocols in backscatter communication

In the conclusion we made the following changes:

Aperture and number of degrees of freedom of our programmable metasurface exceed those of traditional backscatter communication systems by three and a few hundred orders of magnitude, respectively. The resulting wave control enables focusing and multi-channel schemes, which allowed us to demonstrate unprecedented data rates on the order of hundreds of Kbps and communication security – without requiring an active radiofrequency chain, energy and spectral resources to generate a carrier signal. Therefore, the reported MBWC strategy has the potential to fundamentally resolve pressing issues in conventional AWC (energy consumption, spectrum allotment, hardware cost, information security) **and in conventional backscatter communication (low data rates, lack of security)**, notably in the context of green IoT connectivity. We believe that our MBWC strategy provides a fundamentally new view on **the role of programmable metasurfaces in** wireless communication that can impact a wide range of future communication systems at radiofrequencies and beyond.

To summarize, we hope that the revised terminology in combination with our improved clarity regarding the unique advantages of our scheme over the traditional backscatter literature allow the reviewer to appreciate the technological impact of our proposal and to approve the publication of our manuscript.

REVIEWERS' COMMENTS:

Reviewer #2 (Remarks to the Author):

Authors have satisfactorily respond my concern. I would like to recommend the paper for publication.